# Clinical and Laboratory Parameters After Drowning and Diving Accidents and Their Association with Survival

**DOI:** 10.3390/pathophysiology32040065

**Published:** 2025-11-19

**Authors:** Anne Petzold, Jan Dreßler, Anne Schrimpf, André Gries

**Affiliations:** 1Institute of Forensic Medicine, Faculty of Medicine, Leipzig University, 04103 Leipzig, Germany; jan.dressler@medizin.uni-leipzig.de; 2Emergency Department, University Hospital of Leipzig, 04103 Leipzig, Germany; andre.gries@medizin.uni-leipzig.de; 3Institute for General Practice, Faculty of Medicine, Leipzig University, 04109 Leipzig, Germany; anne.schrimpf@medizin.uni-leipzig.de

**Keywords:** drowning, diving accidents, mortality, survival, emergency department, survival prediction, prognostic markers

## Abstract

**Introduction**: The prognosis for patients admitted to emergency departments (ED) after drowning or diving accidents is often uncertain. In this study, we evaluated a range of clinical and laboratory parameters as potential indicators of survival. Many of these markers have previously been investigated in the context of survival prediction in both trauma-related and non-trauma-related clinical scenarios. **Methods**: We conducted a retrospective analysis of 25 patients aged >17 years who were admitted to the ED of the University Hospital Leipzig after drowning or diving accidents between 2012 and 2024. Clinical and laboratory parameters were compared between survivors and non-survivors, with survival defined as discharge from the hospital. **Results**: Of all cases analyzed—comprising 19 drowning and six diving incidents—10 patients (40%) survived, while 15 (60%) did not. Age, sex, or etiology of the accident were not statistically associated with survival. Compared to survivors, non-survivors were significantly more likely to have received prehospital cardiopulmonary resuscitation (CPR; 20% vs. 86.7%) and to have exhibited lower Glasgow Coma Scale scores and lower pH values (7.4 vs. 6.7). They were also more likely to have shown increased levels of lactate (4.3 mmol/L vs. 14.8 mmol/L), CK-MB quotient (9.7% vs. 51.8%), myoglobin (188.9 µg/L vs. 1930.9 µg/L), and blood glucose (6.6 mmol/L vs. 14.3 mmol/L). **Conclusions**: The need for CPR appears to be the most significant risk factor for not surviving a drowning or diving accident. Furthermore, certain laboratory parameters, such as pH and lactate, may provide supportive information regarding the severity of hypoxia and could be cautiously considered as indicators of survival likelihood in these patients. Our findings offer a rationale for future prospective studies, aiming to incorporate additional clinical and biochemical markers and potentially develop new prognostic scoring systems for patients following drowning or diving accidents. This study examines the association between clinical and laboratory parameters and survival in patients following drowning and diving accidents. A total of 25 cases from 2012 to 2024 were retrospectively analyzed. The results showed that patients who required CPR had significantly poorer outcomes. Certain laboratory markers; such as pH and lactate levels; were closely related to survival status in this patient group.

## 1. Introduction

Worldwide, unintentional drowning causes more than 500,000 deaths each year [1]. In Germany, the German Lifesaving Society reported 411 drowning-related deaths in 2024 [2]. According to the German Resuscitation Registry (GRR), drowning accounted for 0.4% (approximately 42 cases) of all out-of-hospital resuscitations in 2023 [3]. In contrast to drowning, there is currently no centralized registry for diving accidents in Germany [4]. However, the German Association of Sports Divers documented 62 diving accidents in Germany in 2018 [5]. Other European studies have also reported relatively low numbers of diving accidents. For example, a Spanish study recorded only 25 cases over a 10-year period [6]. However, the true burden of diving-related incidents remains uncertain, as most patients with clinically diagnosed decompression sickness (DCS) are not systematically followed up, resulting in incomplete and unreliable data on long-term outcomes [4,7,8]. Overall, the lack of a comprehensive database encompassing both fatal and non-fatal drowning and diving incidents has been a major limitation in assessing their full epidemiological and clinical impact [9].

The prognosis for affected patients upon hospital admission is often uncertain and depends on several factors. Diving accidents frequently result in DCS, which is caused by the formation of inert gas bubbles (e.g., nitrogen) during rapid ascent to the surface. According to the S2k guidelines, DCS is categorized based on symptom severity: mild cases present with skin irritation and fatigue, whereas severe cases present with pain and neurological and/or cardiocirculatory symptoms [7,10]. Additionally, pulmonary barotrauma and arterial gas embolism may occur, both of which can be fatal [7,11]. Although several predisposing factors can lead to diving accidents and a generally different pathology, drowning remains the most common forensic cause of death in fatal diving cases, as shown in numerous other studies [11,12,13,14]. Therefore, we chose to group and analyze these two entities together.

According to the current S2k guideline from March 2023 [10], the primary therapeutic intervention in diving accidents is the immediate administration of 100% oxygen, or hyperbaric oxygen therapy (HBOT) in severe cases. The main objective of emergency treatment in both drowning- and diving-related accidents is the rapid management of hypoxia to prevent irreversible neurological injury and loss of function [15]. Airway management, fluid resuscitation, and cardiopulmonary resuscitation (CPR) according to the most recent guidelines may be necessary [7,10].

Upon admission to the emergency department (ED) at a hospital, the prognosis of patients involved in drowning or diving accidents largely depends on the nature of the incident and, in case of drowning, the duration of submersion. Several pathophysiological factors associated with these incidents, such as hypothermia, the diving reflex, and respiratory failure, can lead to bradycardia, myocardial ischemia, and cardiac arrest [16].

Laboratory parameters, such as pH, lactate concentrations, blood glucose level, and cardiac biomarkers (troponin, myoglobin, and CK-MB quotient) may serve as indicators for survival and clinical outcome. For example, alterations in blood glucose have been used to predict ED mortality among trauma patients [17], suggesting potential applicability in assessing outcomes after drowning or diving accidents. Similarly, the dynamics of lactate acidosis have been investigated as a potential predictor of 30-day mortality in non-trauma patients following CPR [18]. Both single and sustained elevations in lactate levels are associated with poor clinical outcomes [19]. Cardiac biomarkers, such as troponin, CK-MB, and myoglobin, remain standard laboratory measures for detecting acute myocardial infarction and ischemia, although their diagnostic and prognostic utility continues to be critically evaluated and debated [20,21].

In our study, we evaluated clinical and laboratory parameters that may be associated with survival outcomes following drowning or diving accidents. Previous studies have examined similar parameters, particularly lactate concentrations, as prognostic indicators of survival in both trauma and non-trauma patients [18,19,22]. The primary objective of this study was to evaluate whether these parameters show comparable associations with survival status in patients presenting in the ED after drowning- or diving-related incidents.

## 2. Methods

### 2.1. Study Design

In a retrospective study, we evaluated patients aged over 17 years who had been involved in drowning or diving accidents and were admitted to the ED of the University Hospital of Leipzig between 1 January 2012 and 31 December 2024.

### 2.2. Data Source and Variables

Patient data were retrieved from the electronic patient management system IS-H (Industry Solutions Healthcare by SAP, version 618) using the relevant ICD-10 codes (International Statistical Classification of Diseases and Related Health Problems, 10th Revision) for drowning: T75.1; decompression sickness: T70.3; and diving accident: W16.

We obtained information on patients’ age, sex, the type of accident they were involved in, and their clinical and laboratory parameters upon admission to the ED (standardized admission laboratory) (see Table 1 for an overview of these parameters). Patients were classified as survivors or non-survivors based on their hospital discharge status, with survival defined as discharge from the hospital and non-survival as in-hospital death. We also evaluated cause of death in the non-survivor group.

### 2.3. Statistical Analyses

All statistical analyses were carried out using IBM SPSS Statistics 29 (Armonk, NY, USA) with a two-sided α level of 0.05. For descriptive statistics, missing values in single variables were considered by presenting absolute values, range, and frequencies as % (n/invalid). Continuous variables were presented as mean (M) ± standard deviation (SD).

Differences between categorical variables were analyzed using Fisher’s exact tests. We specifically analyzed differences between survivors and non-survivors in the following categorial variables: “case” (drowning, diving), “sex” (male, female), “CPR received” (yes, no), and “GCS” (GCS ≤ 8, GCS 9–12, GCS ≥ 13). Estimated effect sizes were reported using Phi (φ).

The distribution of all continuous variables was assessed for normality, performed separately for survivors and non-survivors. The Shapiro–Wilk test indicated that the variables “age”, “body temperature”, “CK-MB quotient”, “pH”, “lactate”, and “blood glucose” followed a normal distribution. Group differences between survivors and non-survivors in these variables were analyzed using univariate analyses of variance (ANOVAs). Estimated effect sizes were reported using partial eta squared (η*_p_*^2^).

The Shapiro–Wilk test indicated that “troponin” and “myoglobin” did not follow a normal distribution. Group differences between survivors and non-survivors in these variables were therefore analyzed using Mann–Whitney U-tests. Effect sizes (Cohen’s *d*) for each test with significant results were calculated.

## 3. Results

### 3.1. Patient Characteristics

During the study period, 25 patients were admitted to the ED and included in the analysis [sex: 17 (68%) male, 8 (32%) female, mean age: 49.3 ± 20.5 years]. Of these patients, 19 (76%) had experienced a drowning accident, while 6 (24%) had been involved in a diving accident. The location of the drowning or diving accident was a lake in 19 cases (76%), other freshwater sources in four cases (16%), a pool in one case (4%), and a bathtub in one case (4%).

In total, 15 patients (60%) did not survive the accident. Following drowning incidents, six patients (31.6%) survived, while 13 did not (68.4%). Following diving accidents, four patients (66.7%) survived and two did not (33.3%). Age, sex, or etiology of the accident were not statistically associated with survival. The detailed characteristics of survivors and non-survivors can be found in Table 2.

### 3.2. Clinical Findings in Survivors and Non-Survivors

In the group of non-survivors, CPR was more commonly administered (86.7%) than in the group of survivors (20%). Of the 15 patients (60%) who received CPR, two (13.3%) survived and 13 (86.7%) did not. There was a significant association between receiving CPR and the survival rate (*p* = 0.002). Of the four patients in whom CPR was performed for up to 10 min, two survived. In contrast, none of the 11 patients who received CPR for more than 10 min survived. When the no-flow time exceeded 5 min (in eight patients receiving CPR, 53.3%) none survived, whereas among the six patients (40%) with a no-flow time of less than 5 min, two survived.

In our sample, the GCS values ranged from 3 to 15, with 18 patients showing severe brain injury (GCS ≤ 8) and 7 patients showing minor neurological impairment (GCS ≥ 13). No patients showed moderate neurological impairment (GCS 9–12). All patients in the non-survivor group exhibited a GCS score of 3, which differed significantly from the scores of survivors (*p* < 0.001).

The mean body temperature across all clinical cases was 34.6 ± 2.9 °C (range 28.0–37.0 °C). In non-survivors, body temperature was lower (33.5 °C) than in survivors (36.1 °C); however, this difference was not statistically significant (*p* = 0.100). A detailed description of the clinical parameters for survivors and non-survivors can be found in Table 2.

Of the 15 non-survivors, forensic autopsy revealed that hypoxic brain injury was the leading cause of death in seven cases (46.7%), followed by heart failure and multi-organ failure in four cases each (26.7%).

### 3.3. Laboratory Findings in Survivors and Non-Survivors

The mean lactate value across all clinical cases was 10.5 ± 8.5 mmol/L (range 0.9–28.0), with values of 4.3 mmol/L in survivors and 14.8 mmol/L in non-survivors (*p* = 0.002; Table 2; Figure 1A).

The mean pH value across all clinical cases was 7.0 ± 0.4 (range 6.3–7.5). Consistent with the lactate levels, there was a significant difference in pH between survivors (pH = 7.4) and non-survivors (pH = 6.7; *p* < 0.001; Table 2; Figure 1B).

The mean blood glucose level across all clinical cases was 11.6 ± 5.0 mmol/L (range 4.9–20.7), with blood glucose levels of 6.6 mmol/L in survivors and 14.3 mmol/L in non-survivors (*p* < 0.001).

The mean values for the heart enzymes across all clinical cases were 49.2 ± 60.2 pg/mL for troponin (range 4.6–211.0), 1253.5 ± 1754.0 µg/L for myoglobin (range 25.0–7010.0), and 35.5 ± 26.4% for CK-MB quotient (range 4.6–86.9). Among survivors versus non-survivors, the respective values were 19.2 pg/mL versus 67.7 pg/mL (*p* = 0.076) for troponin, 188.9 µg/L versus 1930.9 µg/L (*p* < 0.001) for myoglobin, and 9.7% versus 51.8% (*p* < 0.001) for the CK-MB quotient. A detailed description of the laboratory parameters for survivors and non-survivors can be found in Table 2.

## 4. Discussion

This retrospective study provides an overview of clinical and laboratory parameters associated with survival outcomes in patients admitted to the emergency department after drowning or diving accidents. Although the sample size was limited, several patterns emerged. The need for CPR, lower GCS scores, severe metabolic acidosis characterized by low pH and elevated lactate levels, and increased concentrations of myoglobin, CK-MB quotient, and blood glucose on admission were significantly associated with in-hospital mortality. In contrast, no significant associations were observed between survival and sex, age, body temperature, troponin levels, or the type of incident (drowning versus diving). These findings are consistent with prior evidence showing that prolonged cardiac arrest, hypoxia, and severe metabolic acidosis are robust indicators of poor outcomes following cardiac arrest and hypoxic injury of any cause [18,19,22]. Our results therefore corroborate these established relationships and further confirm that these prognostic markers might apply to drowning- and diving-related hypoxia, where data remain comparatively sparse.

Our study found a significant association between receiving out-of-hospital CPR and mortality, indicating that patients requiring CPR after a drowning or diving accident are less likely to survive. This finding aligns with established evidence showing that even brief periods of cardiac and pulmonary dysfunction markedly affect survival and neurological outcomes, irrespective of the cause of out-of-hospital cardiac arrest (OHCA) [26]. The necessity and duration of CPR, along with parameters such as no-flow time and return of spontaneous circulation (ROSC), are well-recognized determinants of survival [4,24,26], although the prognostic value of CPR duration remains debated [27,28]. Given that drowning inherently leads to terminal apnea and cardiac arrest, the strong association between receiving CPR and poor outcomes is coherent. In 2023, the GRR reported ROSC in only 41.4% of OHCA cases, with 24 h survival was below 20% [3]. In our cohort, only two of fifteen patients who received CPR survived; both had CPR durations under 10 min, whereas all eleven patients resuscitated for more than 10 min died. This observation aligns with previous findings that prolonged resuscitation (>20 min) is associated with lowest survival rates, while shorter durations (<10 min) yield the best outcomes [27]. Although our small sample precluded statistical testing for no-flow time, all survivors had a no-flow period of less than 5 min, compared to over 5 min in most non-survivors. This likely reflects longer rescue times from the water. Other studies have found that no-flow time could be a reliable prognostic marker for survival and neurological outcome in patients with OHCA of endogenous origin [29], though our data were insufficient to confirm this.

The leading clinical cause of death among non-survivors was hypoxic brain injury. Previous research has linked neurological outcomes to CPR duration, timing of intervention, and hypothermia management [30]. In drowning and diving incidents, these parameters are difficult to determine, as rescue from the water is often delayed, making hypothermia more likely [31]. This may explain the predominance of hypoxic brain injury as the leading cause of death.

Hypothermia is considered part of the “triad of death”, along with coagulopathy and acidosis [32]. However, it may also exert neuroprotective effects and is used therapeutically to mitigate hypoxic brain injury [15]. In our study, no significant association was observed between body temperature and survival, although non-survivors generally presented with sub-physiological temperatures. Other studies have described correlations between low rectal temperatures and poorer clinical outcomes in both drowning- and non-drowning-related cases [33,34].

The GCS is commonly used in prehospital settings to assess consciousness following traumatic brain injury [25]. In our cohort, all non-survivors had a GCS of ≤8, whereas most survivors presented with a GCS of ≥13. Given that loss of consciousness is a central feature of the pathophysiology of drowning, a strong association between survival and GCS is expected [25]. This relationship has also been consistently demonstrated in studies on OHCA and traumatic injury [24,35].

We found that pathological values in several laboratory markers were associated with survival in drowning and diving accidents. Upon admission, non-survivors exhibited more severe metabolic acidosis (low pH and high lactate levels) than survivors. In this context, acidosis is a direct consequence of prolonged tissue hypoxia and anaerobic metabolism. Similar findings were reported in a previous study which found that lactate was as an early predictor of survival and clinical outcomes in patients receiving resuscitative care [36]. This study also demonstrated a significant association between survival and arterial pH, identifying pH as an independent predictor of survival [36]. Given the well-established inverse relationship between lactate and pH, our result was anticipated [19]. The differences in pH and lactate levels between survivors and non-survivors suggest that these routinely available laboratory markers can serve as rapid indicators for mortality risk upon admission. While these findings are not novel in the broader context of resuscitation medicine, this study confirms their critical prognostic importance within the specific, less frequently studied cohort of drowning and diving-related incidents. In the absence of large, prospective databases for aquatic accidents, such retrospective analyses remain important for characterizing the biochemical and clinical profiles of this rare patient population.

A more contentious finding of our study is the strong association between elevated cardiac biomarkers, specifically CK-MB and myoglobin, and mortality. Both myoglobin and CK-MB are usually used in diagnosing myocardial infarction but can also be significantly elevated due to noncardiac skeletal muscle injury (e.g., rhabdomyolysis) or renal failure [20]. Resuscitation efforts, especially prolonged CPR, are also known to cause elevations in these markers, confounding their interpretation as indicators of primary cardiac injury [37]. This is supported by findings from another study which investigated differences between survivors and non-survivors who had both received CPR, reporting elevated CK-MB ratios without significant differences between groups [36], suggesting that the increase in CK-MB is primarily a consequence of CPR itself rather than an indicator of survival outcome. Therefore, the elevated levels in our non-survivor group, which predominantly consisted of patients requiring prolonged CPR, may primarily reflect the consequences of resuscitation and systemic hypoxia rather than a primary myocardial infarction. Troponin, which is more specific to the heart, did not show a statistically significant association with survival, further underscoring the likely non-cardiac origin of the CK-MB and myoglobin elevations. Rather than proposing these markers as reliable predictors of outcome, our findings should be understood as reflecting the extent of systemic hypoxia and tissue injury in non-survivors and might serve as a preliminary, hypothesis-generating observation that highlights the severe systemic insult experienced by these patients.

We also observed significantly higher blood glucose levels in non-survivors, which highlights the potential relevance of hyperglycemia in the resuscitation phase. Although research on this specific association in drowning and diving accidents is limited, our results are in line with a previous study in trauma patients, showing that elevated blood glucose levels were associated with neurological outcomes and survival [17]. Whether this represents a stress response to the drowning or diving event remains uncertain, but it may warrant further investigation.

Overall, the results of our study indicate that predictors of survival following drowning and diving accidents are broadly comparable to those observed in other clinical contexts, including both traumatic and non-traumatic fatalities [18,38]. Although the severity and pathophysiological presentation of certain parameters may differ in drowning compared with other causes of hospitalization, their implications for survival appear similar, despite the limited sample size precluding definitive conclusions. Drowning and diving incidents, however, pose distinct challenges, such as prolonged rescue times, environmental influences such as water temperature, and extended submersion or no-flow times. These unique circumstances emphasize the need for rapid assessment and initiation of appropriate interventions, including CPR, thermoregulation, HBOT, and other clinical treatments.

This study has several significant limitations that must be acknowledged. The most important one is the small, heterogenic sample size of 25 patients, which severely limits the statistical power and the generalizability of our findings. The retrospective and monocentric design introduces potential biases related to data collection and missing variables. Furthermore, by grouping drowning and diving accident patients, we combined two entities with potentially different underlying pathophysiology, although drowning is often the final pathway in fatal diving incidents. Important prognostic variables such as submersion duration, water temperature, and time to initiation of bystander CPR could not be consistently analyzed due to the retrospective nature of the data. Not all clinical and laboratory parameters were available for every patient, which impeded the application of multivariate analysis to identify the independent effects of these parameters. As a cross-sectional study, outcome classification was limited to in-hospital survival, without long-term neurological follow-up in surviving patients. Additionally, evolving clinical treatment protocols over the study period (2012–2024), along with incomplete data for some variables, further constrain the robustness of the results. It should be noted that most of the laboratory parameters analyzed are nonspecific to drowning- and diving-related accidents. As discussed, the prognostic value of CK-MB and myoglobin is likely confounded by CPR-induced muscle injury and should therefore be interpreted with caution. Consequently, this study should be regarded as preliminary and underscores the need for further research with larger cohorts.

## 5. Conclusions

The primary aim was to determine whether established predictors of survival could also be applied to patients involved in drowning- and diving-related incidents. Our preliminary study supports the established associations between survival outcome and clinical and laboratory parameters, such as CPR requirement, low GCS score, and severe acidosis in patients following drowning or diving accidents. It also adds to the limited body of evidence characterizing the biochemical and clinical profiles of this specific patient population. Although laboratory parameters can be nonspecific, they provide valuable insight into the severity of submersion-related emergencies. In our study, lactate levels were markedly elevated, and pH values correspondingly reduced in non-survivors. However, the observed association between non-specific biomarkers like myoglobin and CK-MB and mortality must be interpreted with caution, as these are likely confounded by the effects of prolonged CPR and systemic injury. As clinical data on diving or drowning accidents remain limited and difficult to obtain, our results might serve as a preliminary, hypothesis-generating observation. There is a clear need for larger, prospective, multi-center registry studies to validate these observations, enable more refined risk stratification, and ultimately improve the management and prognosis of drowning and diving incidents. Furthermore, the development of extended emergency and rescue protocols and follow-up documentation should be encouraged.

## Figures and Tables

**Figure 1 pathophysiology-32-00065-f001:**
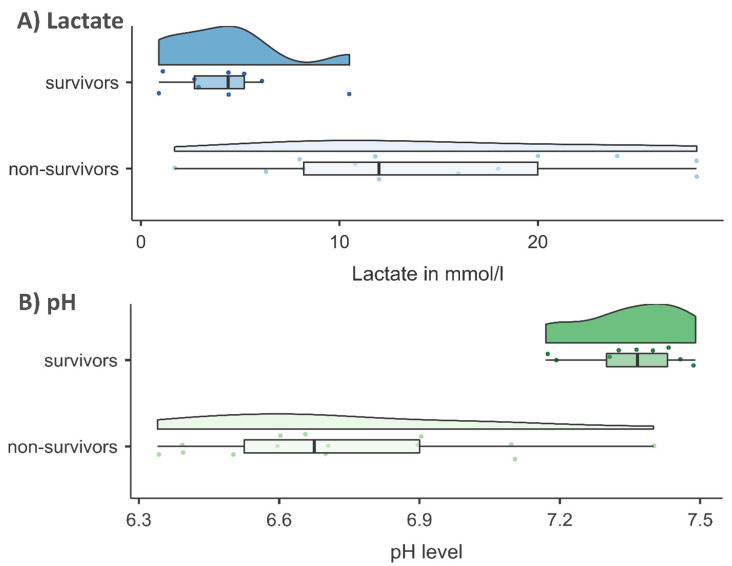
Violin plots of (**A**) lactate in mmol/L and (**B**) pH level in survivors and non-survivors.

**Table 1 pathophysiology-32-00065-t001:** Clinical and laboratory parameters recorded in patients with drowning/diving accidents in the emergency department.

Patient Characteristics	Age [yrs]	Sex [m/f]	Survival [Yes/No]	Drowning/Diving accident
**Clinical parameters on admission**	Initial body temperature [°C] (36–37) ^1^	CPR duration [≤10 min/>10 min]	No-flow time ^2^ [≤5 min/>5 min]	Glasgow Coma Scale (GCS) ^3^ (3–15)
**Laboratory parameters on admission**	Lactate [mmol/L] (0.5–2.2)	pH value (7.36–7.44)	Blood glucose [mmol/L] (4.0–7.8)	Heart enzymes [troponin in pg/mL (<14), CK-MB quotient in % (<6), myoglobin in µg/L (28–72)]

Note. ^1^ values in parentheses () represent the physiological range [23]. ^2^ No-flow time is defined as time interval between onset of out-of-hospital cardiac arrest and start of cardiopulmonary resuscitation [24]. ^3^ While the GCS was originally developed for assessing consciousness levels in traumatic brain injury [25], application of GCS in this study reflects its utility in evaluating neurological function irrespective of underlying etiology.

**Table 2 pathophysiology-32-00065-t002:** Clinical and laboratory parameters in groups of survivors and non-survivors after drowning or diving accidents.

Parameter	Survivor	Non-Survivor	*p*
** *N* **	10	15	
**Patient characteristics**			
**Case [drowning/diving]**	6/4	13/2	χ^2^(1) = 2.339, *p* = 0.175, φ = 0.31
**Sex [m/f]**	7/3	10/5	χ^2^(1) = 0.031, *p* = 1.000, φ = 0.04
**Age [yrs]**	51.6 ± 22.2	47.8 ± 20.0	*F*(1, 23) = 0.199, *p* = 0.660, η*_p_*^2^ = 0.01
**Clinical findings**			
**CPR [yes/no]**	2/8	13/2	χ^2^(1) = 11.111, *p* = 0.002, φ = 0.67
**If yes:**			
**CPR duration [≤10 min, >10 min]**	2/0	2/11	n.a.
**No-flow time [≤5 min, >5 min]**	2/0	4/8	n.a.
**Body temperature [°C]**	36.1 ± 1.0	33.5 ± 3.4	*F*(1, 12) = 3.186, *p* = 0.100, η*_p_*^2^ = 0.21
**GCS**			χ^2^(1) = 14.583, *p* < 0.001, φ = 0.76
**GCS ≤ 8**	3	15	
**GCS 9–12**	0	0	
**GCS ≥ 13**	7	0	
** *Laboratory findings* **			
**Lactate [mmol/L]**	4.3 ± 2.9	14.8 ± 8.4	*F*(1, 20) = 13.135, *p* = 0.002, η*_p_*^2^ = 0.40
**pH**	7.4 ± 0.1	6.7 ± 0.3	*F*(1, 21) = 32.210, *p* < 0.001, η*_p_*^2^ = 0.61
**CK-MB quotient [%]**	9.7 ± 4.5	51.8 ± 20.5	*F*(1, 16) = 28.153, *p* < 0.001, η*_p_*^2^ = 0.64
**Myoglobin [µg/L]**	188.9 ± 190.4	1930.9 ± 1977.4	*U* = 3.000, *p* < 0.001, *d* = 0.76
**Troponin [pg/mL]**	19.2 ± 12.9	67.7 ± 70.6	*U* = 27.000, *p* = 0.076, *d* = 0.40
**Blood** **glucose** **[mmol/L]**	6.6 ± 1.4	14.3 ± 4.1	*F*(1, 21) = 26.298, *p* < 0.001, η*_p_*^2^ = 0.56

Note. Values represent n, mean, and standard deviation. n.a. = *p* value not applicable due to low sample size in the survivor group.

## Data Availability

The datasets used during the current study are available from the corresponding author on reasonable request.

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
