# Peer review of "Clinical and Laboratory Parameters After Drowning and Diving Accidents and Their Association with Survival"

_pathophysiology, 2025, doi:10.3390/pathophysiology32040065_

Round 1
Reviewer 1 Report
Comments and Suggestions for Authors
The manuscript addresses an interesting clinical problem, namely the prognostic value of clinical and laboratory parameters in patients admitted after drowning or diving accidents. While the topic is relevant for emergency and forensic medicine, the current work has significant issues.
The most evident shortcoming is the extremely small sample size of 25 patients collected over a twelve-year period. Such a limited cohort cannot provide statistical evidence, particularly when stratified into survivors and non-survivors or into drowning versus diving accidents. The conclusions drawn from these data are therefore highly speculative and cannot be generalized. The authors nonetheless present their findings with a degree of certainty that is not justified by the evidence, which risks misleading readers.
The methods section describes standard statistical tests. The assumption of normal distribution for certain variables is questionable since the mixture of ANOVAs, Mann–Whitney tests, and Fisher’s exact tests on such small subgroups is methodologically weak. The authors also fail to report confidence intervals, which makes it not possible to judge the robustness of the associations.
The results are presented largely as a list of significant p-values without deeper analysis. The finding that CPR and low pH/lactate levels are associated with mortality is unsurprising. The paper therefore adds little new knowledge. The attempt to extend prognostic value to markers such as CK-MB or myoglobin is problematic, since these biomarkers are non-specific, highly influenced by CPR itself, and already known to lack prognostic reliability in critical care. This limitation is only briefly acknowledged in the discussion, but the authors nevertheless emphasize these results as if they carried major clinical implications.
The discussion suffers from redundancy and overinterpretation. Much space is devoted to restating the obvious (e.g., patients needing CPR fare worse), while the opportunity to discuss the real limitations of the study, such as sample size, case heterogeneity, lack of follow-up for neurological outcomes, and absence of multivariate analysis, is missed. Again, claims regarding the clinical utility of blood sugar, CK-MB, or myoglobin in this context are not substantiated and risk overstating the contribution of the study. The proposal to develop new prognostic scoring systems based on such weak data is commendable and of medico-legal relevance, but premature.
Stylistically, the manuscript requires substantial editing. Tables and figures are not always integrated into the narrative, and some findings are repeated multiple times without adding insight. Also, awkward phrasing and frequent repetition are present in the manuscript. Several sentences are unnecessarily long.
Author Response
Please find our Response to the reviewer in the file added below.

Reviewer 2 Report
Comments and Suggestions for Authors
It should be included if the study has approval from the hospital's ethics committee for access to information obtained through access to medical records.
The study consists of 25 patients admitted to the emergency department after drowning or diving accidents between 2012 and 2024. The number of cases, as the authors acknowledge, is small, making it difficult to draw conclusions. The contrast between the established groups is very limited due to the total sample size. The authors comment that this is one of the several limitations, which is why it is a preliminary study.
The introduction section is well developed. Perhaps the last paragraph on page 2: 'Many factors surrounding…' should be at the beginning of that paragraph to improve the overall understanding.
Information on the methodology followed to obtain the information and the time at which blood was drawn to determine the concentrations of laboratory parameters should be expanded. The concentrations of the different biochemical parameters may be influenced by treatment and support measures in the emergency department.
The CK-MB and myoglobin concentrations obtained are highly nonspecific, as has been commented by numerous authors, and it is difficult to draw conclusions from these measurements. However, troponin levels did not show statistically significant differences when compared between the Survivor and Nonsurvivor groups. As the authors note, cardiopulmonary resuscitation may interfere with the determination of these biochemical markers.
The conclusions discussed are not related to the objectives of the study.
Author Response
Please find our response in the file added below.

Round 2
Reviewer 1 Report
Comments and Suggestions for Authors
Authors solved appropriately all the comments.
Author Response
Thank you for your review. Please see our Graphical Abstract.

Reviewer 2 Report
Comments and Suggestions for Authors
The study has been improved and the authors have followed the reviewers' recommendations. The paper may be published in present form.
Author Response

(The authors gave the same response as above.)
